# Erdmann's Anion—An Inexpensive and Useful Species for the Crystallization of Illicit Drugs after Street Confiscations †

**Matthew R. Wood** [1] , **Sandra Mikhael** [1], **Ivan Bernal** [1,2] **and Roger A. Lalancette** [1,*]

1. Carl A. Olson Memorial Laboratories, Department of Chemistry, Rutgers University, 73 Warren St., Newark, NJ 07102, USA; mwood@co.ocean.nj.us (M.R.W.); tug90124@temple.edu (S.M.); bernalibg@gmail.com (I.B.)
2. Molecular Sciences Institute, School of Chemistry, University of the Witwatersrand, Private Bag 3, Johannesburg 2050, South Africa
* Correspondence: roger.lalancette@gmail.com; Tel.: +1-973-353-5646
† In Memorian: Howard D. Flack (1943–2017)—he, and his work, will be remembered.

**Abstract:** Erdmann's anion [1,6-diammino tetranitrocobaltate(III)] is useful in the isolation and crystallization of recently confiscated street drugs needing to be identified and catalogued. The protonated form of such drugs forms excellent crystals with that anion; moreover, Erdmann's salts are considerably less expensive than the classically used $AuCl_4^-$ anion to isolate them, while preparation of high-quality crystals is equally easy in both cases. We describe the preparation and structures of the $K^+CoH_6N_6O_8^-$ and $NH_4^+CoH_6N_7O_8^-$, salts of Erdmann's. In addition, herein are described the preparations of this anion's salts with cocaine ($C_{17}H_{28}CoN_7O_{12}$), with methamphetamine ($C_{10}H_{22}CoN_7O_8$), and with methylone ($C_{22}H_{34}CoN_8O_{14}$), whose preparation and stereochemistry had been characterized by the old $AuCl_4^-$ salts methodology. For all species in this report, the space groups and cell constants were determined at 296 and 100 K, looking for possible thermally induced polymorphism—none was found. Since the structures were essentially identical at the two temperatures studied, we discuss only the 100 K results. Complete spheres of data accessible to a Bruker ApexII diffractometer with Cu–Kα radiation, λ = 1.54178 Å, were recorded and used in the refinements. Using the refined single crystal structural data for the street drugs, we computed their X-ray powder diffraction patterns, which are beneficial as quick identification standards in law enforcement work.

**Keywords:** Flack test; Erdmann's anion; bath salts; street drugs; cocaine; methamphetamine; methylone; π–π interactions; racemic mimics; kryptoracemic crystallization



## 1. Introduction

Notes: (a) Erdmann's salt should not be confused with Erdmann's reagent (sulfuric acid containing dilute nitric acid), which has been used as an alkaloid color test [1]. (b) It also should not be confused with the cis–diamino (1,2-diamino) derivative that was described by Shintani, et al. [2]. (c) For the reader's convenience, the six letter acronyms used in the references provide easy access to the Cambridge Crystallographic Database [3] information and CIF documents.

In collaboration with the Ocean County Sheriff's Office Forensic Science Laboratory (NJ, USA), we have been engaged in studies of the nature of the street drugs commonly known as bath salts [4,5], the addictive principle of which are positively charged amino species, per se, or have been converted into hydrohalides (Cl⁻ or Br⁻, or mixtures thereof) in order to make them water soluble. Some of the samples used were from police seizures, which in most cases are of unknown provenance. Because an effective method of isolating and identifying them has, traditionally, been to crystallize them as salts using the expensive $AuCl_4^-$ anion, we decided to find alternative, inexpensive anions, which would be simple to make even by our first-year chemistry major or nonmajor, students. Those salts should provide equally good, hopefully better, microscopic and X-ray diffraction quality crystals

with those of the traditional gold anion samples. Given that all of the street drugs are amines, it is not difficult to assume that, in cationic form, they will readily interact, via hydrogen bonding, with moieties that can act as proton donors–acceptors.

Since a number of these drugs contain oxygen moieties that can act as bases to proton donors, an ideal crystallization partner would be one that can function equally well as either an acid or a base. Such a reagent is Erdmann's anion, which is simple to prepare in multigram quantities at a very low cost and can act both as a proton acceptor and as a proton donor to various cationic drugs. Representative samples (cocaine, methamphetamine, methylone) were selected in order to demonstrate the practical use of the reagent. They were crystallized as Erdmann's salts by the addition of a 5% aqueous solution of either ammonium or potassium Erdmann's anion and a few milligrams of the target drug compound. As the potassium salt, Erdmann's salt was first described in 1866 [6]; later, Jørgensen improved the synthesis of the ammonium salt [7]. The crystal structure of $K[Co(NH_3)_2(NO_2)_4)]$ was initially determined at room temperature by X-ray diffraction using $FeK_\alpha$ radiation ($\lambda = 1.937\text{Å}$) in 1956 [8]. Here, we describe the crystal structures of both the potassium and ammonium salts at 100K using complete spheres of data and give a detailed description of the structures of complexes of three cationic drugs with Erdmann's anion.

## 2. Materials and Methods

Note: Origin of the drugs used in this study: Cocaine (**3**) and methylone (**5**) were obtained from drug seizures. Methamphetamine (**4**) was of pharmaceutical grade (enantiopure), purchased to set up a standard for the forensics laboratory. All other chemicals were of analytical reagent grade and were obtained from Sigma Aldrich (St. Louis, MO, USA), Fisher Scientific (Waltham, MA, USA), or VWR (Radnor, PA, USA) and used without purification. Any law enforcement seizures were of unknown provenance but were characterized by GC/MS analysis.

### 2.1. Syntheses and Crystallization

2.1.1. Syntheses of (**1**) and (**2**)

In order to have a common source of this reagent (Erdmann's anion), it was prepared in a large scale as follows: The potassium salt of Erdmann's anion $K[Co(NH_3)_2(NO_2)_4]$, complex (**1**) (MW = 316.12 g/mol) was prepared by weighing 40.0 g of $CoCl_2 \cdot 6H_2O$ (MW = 237.93 g/mol) (0.168 moles) dissolved in 100 mL of distilled water with stirring. In a separate beaker, 60.0 g $NaNO_2$ (MW = 69.01 g/mol) (0.869 moles) and 35.0 g $NH_4Cl$ (MW = 53.492 g/mol) (0.654 moles) were dissolved in 288 mL of distilled water with stirring and slight heating. This second solution was filtered through a glass frit filter. To the second solution, 12 mL (0.180 moles) of "fresh" conc. $NH_4OH$ (15 M) was added with stirring. Both solutions were combined in a side-arm flask fitted with a rubber stopper and a glass tube (1 cm in diameter) to allow air to be drawn into the mixture. Air was bubbled vigorously through the mixture for 90 min. To the mixture was added 30g KCl (MW = 74.55 g/mol) (0.402 moles), after which it turned from brown to brownish red. The product was transferred to an evaporating dish, where it was left for 2–3 days. It yielded a yellow–brown precipitate and a red–orange liquid. The solid was filtered using glass frit filter, and the precipitate was dissolved in 300 mL of distilled $H_2O$ at 60 °C. After 2 min, the brown solution was filtered through a glass frit filter and then cooled in an ice bath. The resulting crystals were recovered using a glass frit filter, dissolved in 300 mL of hot water, and allowed to crystallize, yielding 28.4 g (53% yield based on Co). For the ammonium salt, complex (**2**), in a different preparation, 21.5 g $NH_4Cl$ were added at the end of the procedure, instead of KCl.

2.1.2. Preparation of the Drug Crystals

Complex (**3**): A sample of a few milligrams of crystalline cocaine·HCl was dissolved on a glass slide in $H_2O$, and a single drop of a 5% Erdmann's potassium salt solution in

water was added. Crystals of cocaine–trans–diamino–tetranitrocobaltiate(III) began to form as yellow needles through slow evaporative condensation at room temperature. A suitable crystal was chosen for single crystal X-ray analysis.

Complex (**4**): Several milligrams of crystalline methamphetamine·HCl were reacted with a drop of the previously prepared 5% solution of potassium Erdmann's salt on a pre-cleaned microscope slide. Yellow rods precipitated from solution and were allowed to grow at room temperature until they reached a size necessary for X-ray diffraction.

Complex (**5**): The synthetic cathinone (methylone) was also crystallized using the potassium Erdmann's salt reagent. A few crystals of crystalline methylone·HCl in water were mixed on a glass slide with the Erdmann's salt test reagent, and small yellow rods quickly grew out of the solution. A sample suitable for the single crystal X-ray diffraction experiment was chosen for analysis.

The identities of all three illicit drug specimens were previously confirmed by standard gas chromatography–mass spectrometry practices at the Ocean County Sheriff's Office Forensic Science laboratory.

### 2.2. Crystallographic Studies

Each of the crystals (**1**–**5**) was mounted on a Cryoloop using Paratone–N and subsequently mounted on a Bruker Smart ApexII diffractometer. Complete spheres of data were recorded at 100 K using Cu–K$\alpha$ radiation, $\lambda$ = 1.54178 Å. Data processing, Lorentz polarization, and face-indexed numerical absorption corrections were performed using SAINT, APEX, and SADABS computer programs [9–11]. The structures were all solved by direct methods and refined by full matrix least squares methods on $F^2$ using the SHELXTL V6.14 program package. All nonhydrogen atoms were refined with anisotropic displacement parameters; all of the H atoms were found in difference electron density maps. The methylene, methine, aromatic, and amine H atoms were placed in geometrically idealized positions and constrained to ride on their parent C atoms, with C–H = 0.99, 1.00, and 0.95 Å, respectively; the H atoms of the nitrogen and oxygen atoms were refined positionally, and their thermal parameters were fixed to be $1.2U_{iso}N$ and $1.5U_{iso}O$, respectively. For (**1**) and (**2**), structural and refinement parameters and the CCDC deposition numbers can be found in Table 1; for (**3**–**5**), the parameters and CCDC numbers are found in Table 2. The hydrogen bonding results are all found in Tables T1–T5 in the Supplementary Information.

**Table 1.** X-ray Experimental Details for the K$^+$ and NH$_4^+$ Erdmann's Salts.

| | **Crystal Data** | |
|---|---|---|
| | (**1**) = Potassium salt | (**2**) = Ammonium salt |
| Chemical formula | $CoH_6KN_6O_8$ | $CoH_{10}N_7O_8$ |
| $M_r$ | 316.14 | 295.08 |
| Crystal system, space group | Orthorhombic, $P2_12_12_1$ | Orthorhombic, $P2_12_12_1$ |
| Temperature (K) | 100 | 100 |
| $a$, $b$, $c$ (Å) | 6.6678(4), 11.1459(7), 12.7205(9) | 6.6760(2), 11.3965(4), 12.7830(4) |
| $\alpha$, $\beta$, $\gamma$ (°) | 90., 90., 90. | 90., 90., 90. |
| $V$ (Å$^3$) | 945.37(11) | 972.57(5) |
| $Z$, $Z'$ | 4, 1 | 4, 1 |
| Radiation type | Cu $K\alpha$ | Cu $K\alpha$ |
| $\mu$ (mm$^{-1}$) | 18.733 | 14.415 |
| Crystal size (mm) | 0.186 × 0.230 × 0.351 | 0.100 × 0.131 × 0.152 |

**Table 1.** *Cont.*

| **Data Collection** | | |
|---|---|---|
| Diffractometer | Bruker APEX2 | Bruker APEX2 |
| Absorption correction | numerical | numerical |
| $T_{min}$, $T_{max}$ | 0.053, 0.175 | 0.210, 0.409 |
| No. of measured, independent, and observed [$I > 2\sigma(I)$] reflections | 7865, 1473, 1459 | 8624, 1632, 1569 |
| $R_{int}$ | 0.034 | 0.027 |
| $(\sin\theta/\lambda)_{max}$ ($\text{Å}^{-1}$) | 0.610 | 0.618 |
| **Refinement** | | |
| $R[F > 2\sigma(F)]$, $wR(F)$, $S$ | 0.019, 0.047, 1.06 | 0.021, 0.048, 1.02 |
| No. of refl., params., restraints | 1473, 164, 0 | 1632, 175, 4 |
| H−atom treatment | refxyz | refxyz |
| *Flack parameter* | −0.011(5) | 0.030(4) |
| CCDC number | 2047594 | 2047595 |

**Table 2.** X-ray Experimental Details for the Three Drug Complexes with Erdmann's Anion.

| **Crystal Data** | | | |
|---|---|---|---|
| | (**3**) = cocaine salt | (**4**) = methamphetamine salt | (**5**) = methylone salt |
| Chemical formula | $C_{17}H_{28}CoN_7O_{12}$ | $C_{10}H_{22}CoN_7O_8$ | $C_{22}H_{34}CoN_8O_{14}$ |
| $M_r$ | 581.39 | 427.27 | 693.50 |
| Crystal system, space group | Triclinic, $P1$ | Monoclinic, $P2_1$ | Triclinic, $P-1$ |
| Temperature (K) | 100 | 100 | 100 |
| $a$, $b$, $c$ (Å) | 6.2403(3), 11.0319(4), 18.9421(7) | 6.3873(2), 13.0182(3), 21.6772(5) | 7.0437(4), 10.3155(7), 10.7697(7) |
| $\alpha$, $\beta$, $\gamma$ (°) | 106.450(2), 93.831(2), 92.655(2) | 90., 94.6700(17), 90. | 90.712(5), 106.330(4), 107.985(4) |
| $V$ ($\text{Å}^3$) | 1244.94(9) | 1796.50(8) | 710.02(8) |
| $Z$, $Z'$ | 2, 2 | 4, 2 | 1, 1 |
| Radiation type | Cu $K\alpha$ | Cu $K\alpha$ | Cu $K\alpha$ |
| $\mu$ ($\text{mm}^{-1}$) | 6.07 | 8.01 | 5.50 |
| Crystal size (mm) | 0.04 × 0.09 × 0.23 | 0.06 × 0.10 × 0.50 | 0.11 × 0.18 × 0.65 |
| **Data Collection** | | | |
| Diffractometer | Bruker APEX2 | Bruker APEX2 | Bruker APEX2 |
| Absorption correction | numerical | numerical | numerical |
| $T_{min}$, $T_{max}$ | 0.347, 0.772 | 0.440, 0.619 | 0.220, 0.682 |
| No. of measured, independent, and observed [$I > 2\sigma(I)$] refl. | 9894, 5432, 4513 | 15772, 5875, 4066 | 6733, 1302, 1100 |
| $R_{int}$ | 0.030 | 0.073 | 0.029 |
| $(\sin\theta/\lambda)_{max}$ ($\text{Å}^{-1}$) | 0.607 | 0.610 | 0.497 |
| **Refinement** | | | |
| $R[F > 2\sigma(F)]$, $wR(F)$, $S$ | 0.038, 0.087, 0.95 | 0.057, 0.073, 0.90 | 0.079, 0.170, 1.06 |
| No. of refl., params., restraints | 5432, 675, 3 | 5875, 471, 1 | 1302, 209, 24 |
| H−atom treatment | mixed | mixed | constr |
| *Flack parameter* | −0.003(4) | 0.024(6) | − |
| $\Delta\rho_{max}$, $\Delta\rho_{min}$ (e $\text{Å}^{-3}$) | 0.43, −0.39 | 0.65, −0.48 | 0.77, −0.43 |
| CCDC number | 2042087 | 2042090 | 2042091 |

### 3. Results

*3.1. Structures* (**1**) *and* (**2**)

3.1.1. The Potassium Salt of Erdmann's Anion (**1**)

Since the potassium and ammonium salts are isomorphous and isostructural, only the packing diagram of the potassium salt (**1**) is displayed in Figure 1.

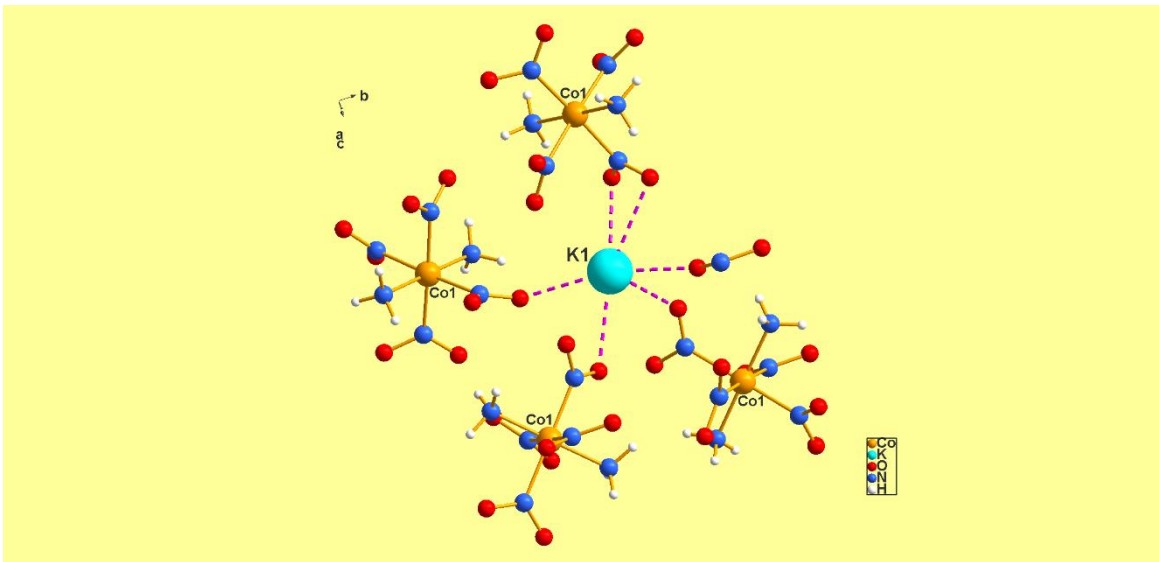

**Figure 1.** The surroundings around the potassium cation present in (**1**). To avoid cluttering, not all of the bonded interactions are shown here. In the ammonium salt, the nitrogen is located exactly at the site of the potassium shown above, but linkages between cation and anion are via $NH_4^+$ hydrogens and the $-NO_2^-$ oxygens on the anion.

Figures 2 and 3, below, display a segment of the packing of the cations and anions in the ammonium salt. As mentioned above (Figure 1's caption), stereochemical information for the ammonium and potassium salts are interchangeable. Note that Figures 2 and 3 are, respectively, *c* and *a* projections, chosen on purpose to display the packing from different angles.

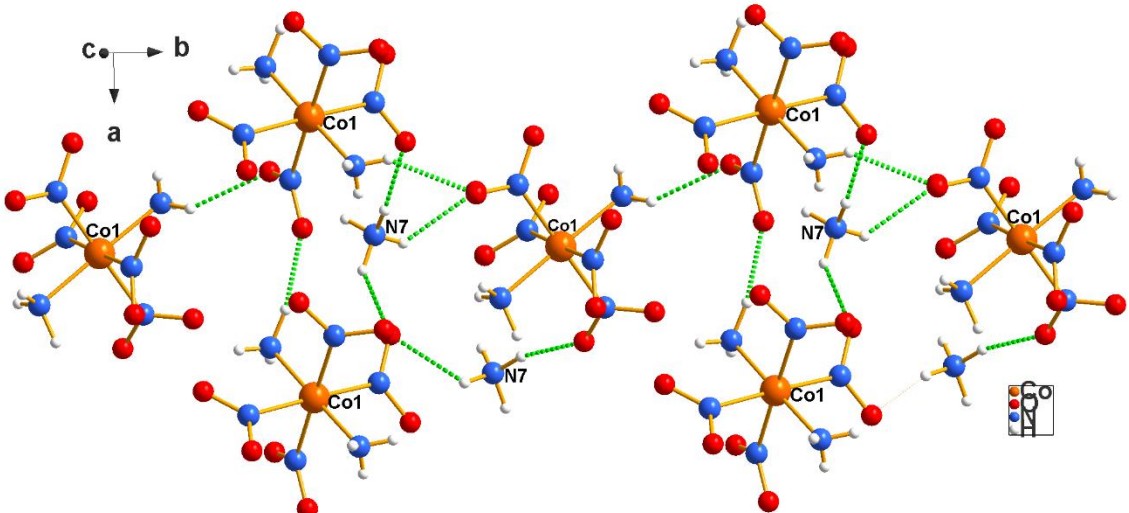

**Figure 2.** The anions form rows parallel to the *b*-axis in the structure of the ammonium salt (**2**), which are linked by the ammonium cations. Identical rows above and below the one shown here continue ad infinitum. For clarity, the complexity of cationic–anionic interactions present is minimally illustrated here. Figures 2 and 3 also illustrate the amphoteric nature of Erdmann's anion, which is the origin of its usefulness as a co-crystallization agent.

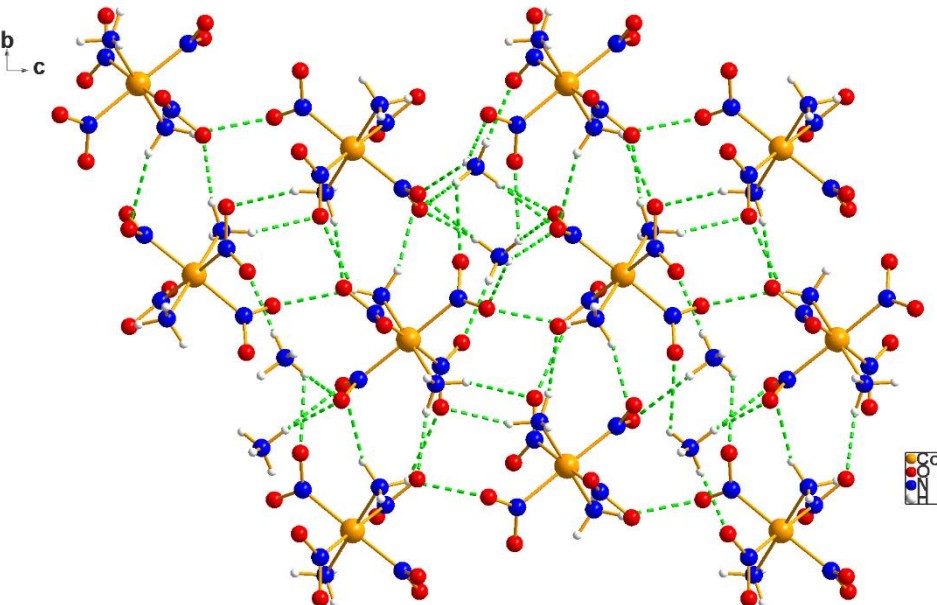

**Figure 3.** In this complicated diagram, the dotted lines define the hydrogen bonds in the ammonium salt of Erdmann's anion (**2**) in which the anions are linked, not only to the ammonium cations, but also to one another.

3.1.2. Structure of the Ammonium Salt of Erdmann's Anion (**2**)

In order to show what a versatile and powerful hydrogen-bonding moiety Erdmann's anion is, we present in Figure 3 the *a*-projection of the packing diagram of its ammonium salt.

In Figure 3, many hydrogen bonds were omitted, because they either (a) clutter the picture or (b) point up or down or both. Note that the anions engage into hydrogen-bonded interactions while acting both as acids (via their $-NH_3$ ligands) or bases (via $-NO_2$ oxygens). In fact, it is just such a versatility that makes Erdmann's anion so attractive for the purification and crystallization of street drugs, which are often contaminated or adulterated for maximizing street profit. Thus, efficient precipitating counter anions serve the dual role of purifying the adulterated material and of providing high quality crystals for X-ray analysis.

*3.2. Structures of the Erdmann's Complexes with Various Street Drugs* (**3**), (**4**), (**5**)

The structures of cocaine, methamphetamine, and methylone, forensically important drugs, were determined by precipitation with the anion of Erdmann's salt, followed by single crystal X-ray diffraction analyses. The former two ((**3**) and (**4**)) crystallize in Sohncke space groups, *P*1 and *P*$2_1$, respectively; thus, their absolute configurations were determined via the Flack Parameter test (see Table 2 and the deposited CIF files for details). The Erdmann's derivative of methylone (**5**) crystallized as a racemate in space group *P*$-1$. In all three cases, the Erdmann's anion from either the potassium or ammonium salt readily replaced the original anion (either $Cl^-$ or $Br^-$) when a few milligrams of the drug compound were reacted with one drop of 5% aqueous solution of Erdmann's anion on a microscope slide. The resulting precipitates are the salts formed by the protonated drug cation and the Erdmann's anion, in which the four $NO_2$ groups and the two *trans*–$NH_3$ groups act as good bases and acids.

A Caveat: It is possible, and sometimes likely, that a crystalline sample, such as methylone, prepared as described above, may give a Flack Parameter value close to 1.0 or 0.0 [12,13], suggesting a pure chiral substance is present despite the fact that this street drug is a manmade racemate. Such result would be due to (a) crystallization in a Sohncke space group as a result of packing, as in the case of $NaClO_3$ or sodium uranyl acetate (both space group *P*$2_1$3), or (b), if Z' = 2.0, and a pair of *near-racemic* molecules caused by small

differences in dissymmetry of flexible fragments caused by packing forces; in that case, the space group may be Sohncke, and the molecules crystallize as kryptoracemates. (For a discussion of the concept of kryptoracemic crystallization, see [14–16]). Additionally, the crystalline material may simply be a case of conglomerate crystallization with Z′ = 2—a widely known phenomenon since Pasteur's day. In all cases, additional measurements, such as CD (circular dichroism) in the solution, etc., would have to be made to correctly interpret the results.

### 3.2.1. Erdmann's Salt of Cocaine, $C_{17}H_{28}CoN_7O_{12}$ (**3**)

The Erdmann's salt of cocaine, $C_{17}H_{28}CoN_7O_{12}$ (**3**), crystallizes in the triclinic Sohncke space group *P*1, with two cocaine cations and two Erdmann's anions in the asymmetric unit (Figure 4). That the space group is *P*1, and not *P*−1 is guaranteed by the fact that the sample is a natural product and that the Flack Parameter test (−0.003(4)) verifies such is the case (see Table 2). There is an intramolecular hydrogen bond in each cation from the quaternary N to the carbonyl O of the methoxy carbonyl moiety: N13–H13 ⋯ O19 is 2.838(7) Å and N14–H14 ⋯ O23 is 2.805(7) Å. One of the cations has an H bond to an O atom on a nitro group on Co1 [N14–H14 ⋯ O8] = 3.052(7) Å. There also exists an H bond from the nearest Erdmann's anion to the ketone [O23] to the cation N6–H6 ⋯ O23[x + 1, y, z] = 3.103(7) Å.

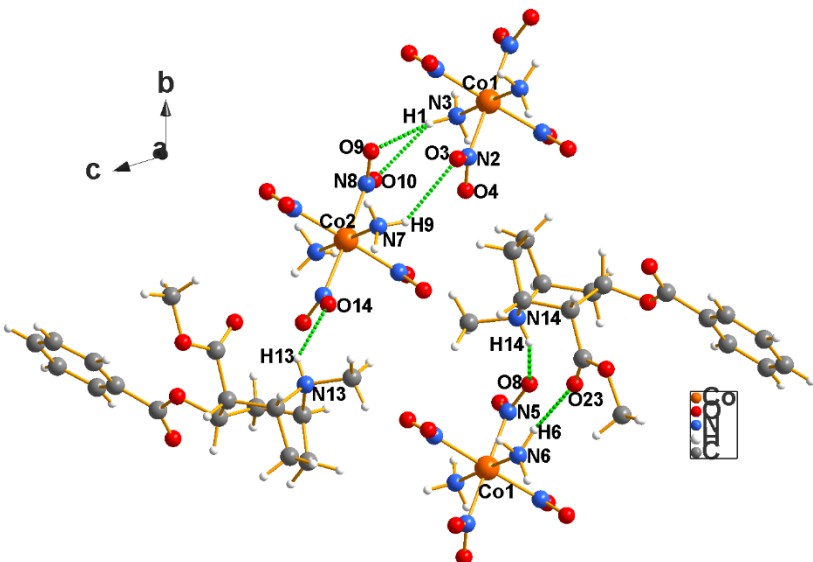

**Figure 4.** The interactions between the cations and anions and cations among themselves in the cocaine–Erdmann's salt (**3**). Note that the –NH$_3$ ligand to Co1 [N6] acts as an acid toward the—C=O oxygen base of the drug [O23], while the drug's ammonium hydrogen atoms [N13] and [N14] act as acids to the –NO$_2$ oxygen atoms [O14 and O8] of the anionic ligands.

### 3.2.2. The Methamphetamine–Erdmann's Complex (**4**)

The Erdmann's salt of pharmaceutical grade enantiopure methamphetamine, $C_{10}H_{22}Co N_7O_8$, crystallizes in the monoclinic space group $P2_1$ with Z = 4 and Z′ = 2. Had the sample been a chiraly mixed, manmade sample, these crystals would constitute a case of conglomerate crystallization, because C1 (from cation 1) and C11 (from cation 2) are both (S) (see Figure 5 below). This is a case in which the Flack Parameter test [13,14] would be of great value to the authorities as a warning of the presence of a meth lab—a nontrivially useful datum for enforcing institutions.

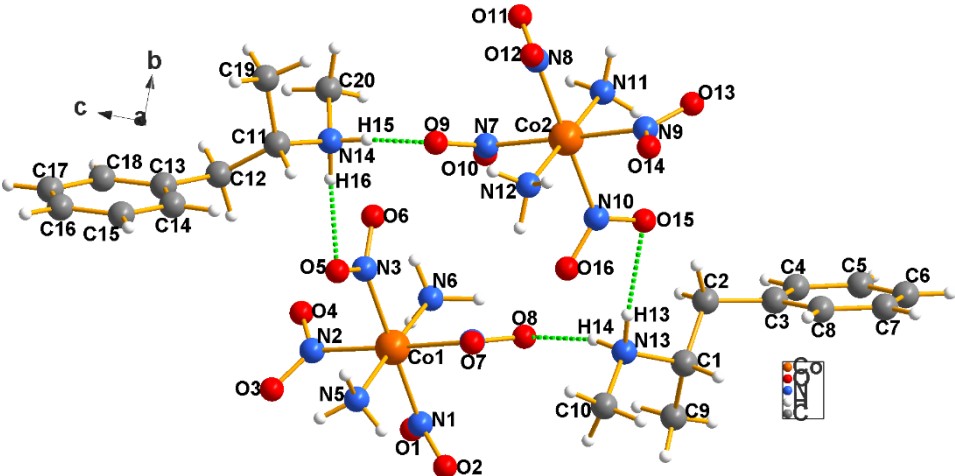

**Figure 5.** There are two independent cation–anion pairs in the asymmetric unit of the methamphetamine specimen we examined (**4**). Note that the cations and anions are linked largely by the NH$_2$ moieties of both cations [N13 and N14], given that the drug has no oxygen atoms of its own. Thus, the hydrogen bonding network is not as robust as it was in the case of the cocaine (see Figure 4 above and Table T4 in the Supplementary Materials). Nonetheless, the fact that the entire lattice is strongly hydrogen bonded leads to crystals of very fine quality.

In Figure 5, one of the Erdmann's anions, with the central metal Co1, makes a hydrogen bond with both proton atoms on N13 of one of the methamphetamine cations in the asymmetric unit [N13–H14 ⋯ O8 = 2.883(8)] and [N13–H14 ⋯ O7 = 3.207(8)] Å. Moreover, there are two other hydrogen bonds from N13 to O15 and O16: N13–H13 ⋯ O15[x + 1, y, z] = 3.070(9) and N13–H13 ⋯ O16[x + 1, y, z] = 3.077(8) Å, and one from N14–H15 ⋯ O9[x, y, z − 1] = 2.851(8) Å. The second anion makes similar H bonds to a symmetry−related cation N14-H16 ⋯ O6[x − 1, y, z − 1] = 3.182(9) and N14-H16 ⋯ O5[x − 1, y, z − 1] = 2.921(8) Å.

### 3.2.3. The Methylone–Erdmann's Complex (**5**)

Methylone, C$_{22}$H$_{34}$CoN$_8$O$_{14}$ (**5**), also forms attractive crystalline lattices with Erdmann's anions, which are useful for its detection and examination; its packing is shown in Figure 6.

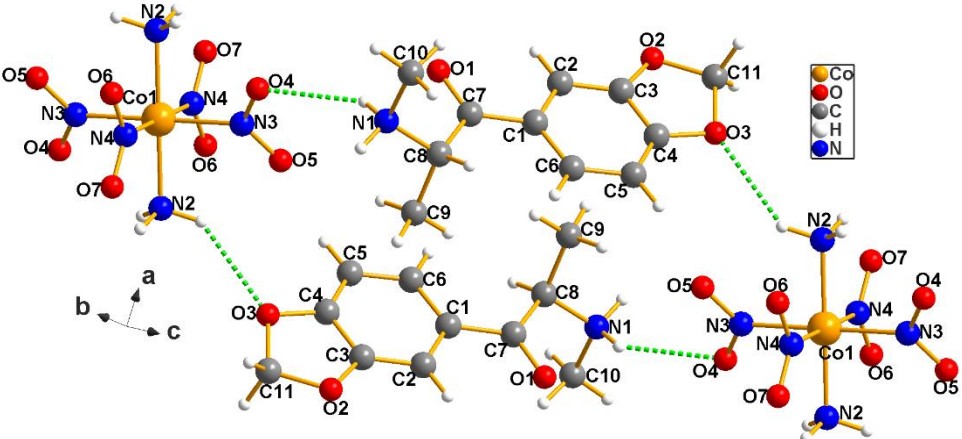

**Figure 6.** Our crystals contain racemic pairs of the drug methylone (**5**), indicating that it is a manmade product, a synthesized analog of cathinone, a stimulant found in *Catha edulis.* Again, note the robust hydrogen bonding network present, in which the anion is displaying its amphoteric nature by linking the equally amphoteric cations. That feature is absent in the case of the classically used AuCl$_4^-$ anions, which are also considerably more expensive.

Methylone crystallizes with Erdmann's anion in *P*−1 triclinic space group (**5**). A pair of symmetry-related anions are joined across the inversion center of the unit cell by a hydrogen bond from N2–H4 to O3[1 − x, 2 − y,−z] = 3.28(1) Å. Additionally, the O4 atoms of the symmetric pair are joined by hydrogen bonds: N1–H1 ··· O4[1 − x, 1 − y,1 − z] = 3.16(1) Å. See Table T5 for bond distances and angles. Figure 6, above, shows the asymmetric unit with an additional symmetry–related anion and cation [−x, −y, z−] present to show the hydrogen bonds and the close contacts and to demonstrate the infinite propagation of anions these close contacts allow.

### 3.3. Packing Considerations in the Three Drug Complexes

Methylone (**5**) crystallizes in *P*−1, with Z′ = 1; thus, no special comments are needed in this case, as is obvious from Figure 6 and comments above. The complexes with cocaine and methamphetamine, however, deserve considerably more careful examination, as illustrated in what follows:

### 3.3.1. Overlay Diagrams of the Drug Fragments for the Complexes with Cocaine (**3**) and with Methamphetamine (**4**)

Given that cocaine and methamphetamine crystallize with Z′ = 2, it was interesting to inquire in what way the two independent components differ; therefore, we resorted to the MERCURY routine of CSD [3]. The resulting overlay Figures 7 and 8 were created with *DIAMOND* [17]. Cocaine (**3**) is shown below:

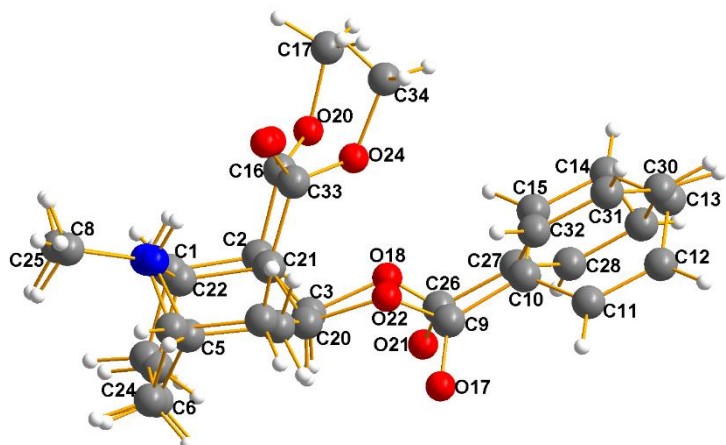

**Figure 7.** Cocaine–Erdmann complex (**3**). This is an overlay of the cationic cocaine molecule2 onto molecule1. The space group is *P*2₁2₁2₁, and Z′ = 2. The program MERCURY [3] was used to overlay cation 2 onto cation 1; then, the fit was optimized. The result displayed above amply justifies the need for Z′ = 2, given the significant differences in torsional angles observed.

This is a simple case of a pure optically active natural product crystallizing in a Sohncke space group with Z′ = 2 because the two molecules are stereochemically flexible and, upon crystallizing, they pack more densely this way. The Flack Parameter [12,13] properly recognizes this, given the fact that the value is −0.003(4). However, there is more to this packing mode, which will be elaborated upon in the section on Racemic Mimics.

Next, we consider the case of methamphetamine (**4**):

Again, as in the case of cocaine, the sample of methamphetamine was known to be chiraly pure, (since it was purchased as a standard material). Therefore, the same comments concerning racemic mimics apply in this case, and relevant comments will be made next.

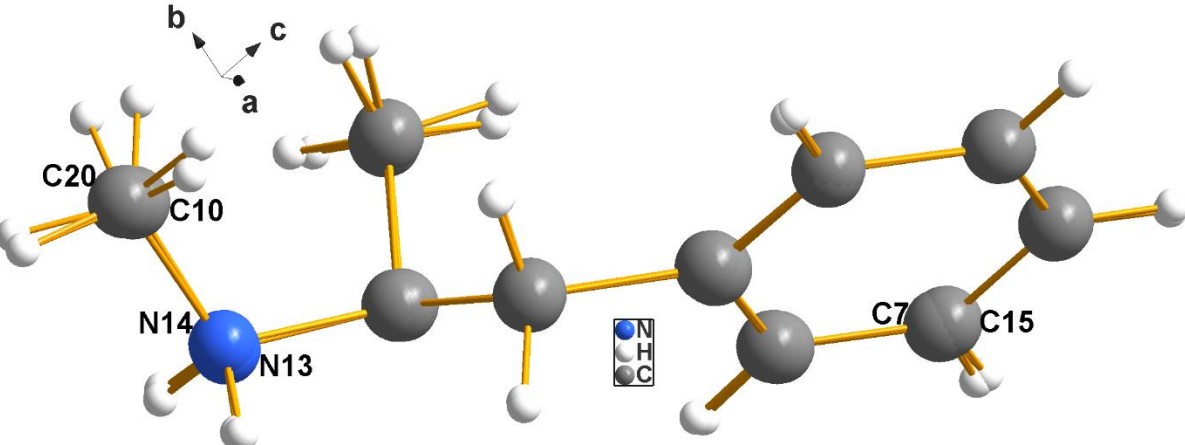

**Figure 8.** Methamphetamine–Erdmann complex (**4**). The overlay here is nearly perfect; the only nonhydrogen atoms that are barely separated enough to discern are shown above. As in the case of the cocaine complex, MERCURY was used to overlay cation 2 onto cation 1, and the fit was optimized. The few labels shown are for those atoms for which the fit was poor enough to allow the observer to note the presence of both atoms.

### 3.3.2. Racemic Mimics

Historically, it appears that an awareness of the existence of this type of crystalline material was first published in papers by a) Furberg and Hassel, who studied the crystal structure of phenyl glyceric acid slowly grown from water [18]; b) Schouwstra, who studied crystals of DL–methylsuccinic acid grown by sublimation [19] and from water solution [20]; and c) Mostad, who examined o–tyrosine crystals grown from methanol containing small amounts of ammonia to increase its solubility [21]. In all those cases, crystals of the racemate and of the optically pure material crystallized with identical cell constants; this leads to values of Z′ = 1 for the racemic samples and Z′ = 2 for the pure enantiomorphs.

[Caveat: because some of those lattices contained racemic pairs and had Z′ = 2.0, the authors of those days [16–19] labeled them racemates. In fact, the proper term today would be kryptoracemates, but because we do not want, at this stage, to branch out into that topic, a brief but suitable discussion of this issue is given in Supplementary Materials 2, below. We thank the referee for bringing this issue to our attention.]

Given that the two lattices (kryptoracemates and Sohncke space groups), Furberg and Hassel [18] asked, "why," and, "how?" In a remarkably clear and simple answer, they indicated that the pure chiral material seemed to crystallize "*as if a twin resembling in its packing that of the true racemate*": in other words, as a "racemic twin"; thus, the name *Racemic Mimics* that later evolved. They also proposed that substances containing flexible (dissymmetric) fragments whose torsional barriers were low would make ideal candidates for the existence of such a phenomenon, and they documented additional cases [18].

(The overlay diagrams shown in this document show the extent to which torsional differences are associated with the observed Z′ value of 2.0). That was a remarkably advanced concept for its day and happens to conform to what we describe in our presentation, since we have two cases of racemic mimics in the cases of the cocaine derivative and of the methamphetamine derivative of Erdmann's salts. For readers interested in more extended commentary on this and related topics, we recommend Herbstein's authoritative compendium [22].

a.    The Case of Cocaine (**3**)

Figure 9 shows the asymmetric unit for the structure of the cocaine-Erdmann's complex.

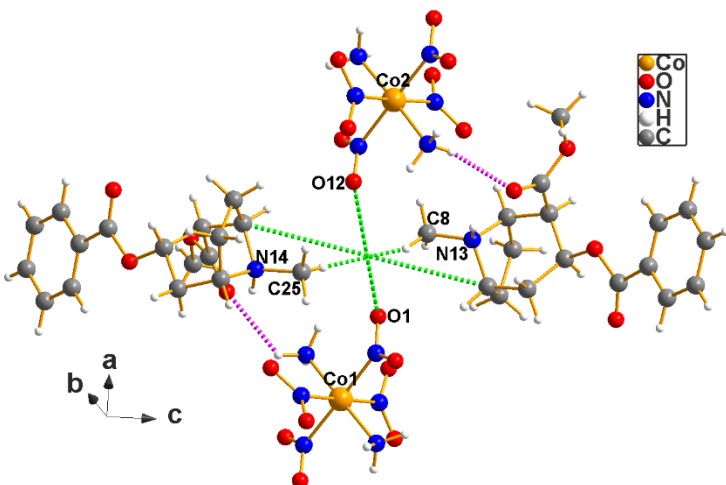

**Figure 9.** The center of mass (0.4741, 0.4173, 0.4689) of the cocaine–Erdmann lattice is located very near to $\frac{1}{2}$, $\frac{1}{2}$, $\frac{1}{2}$, but in *P*1, the origin is totally arbitrary, which renders the issue moot for this case. Note, however, that is not the case for methamphetamine (see Figure 10, next).

b.    The Case of Methamphetamine (**4**)

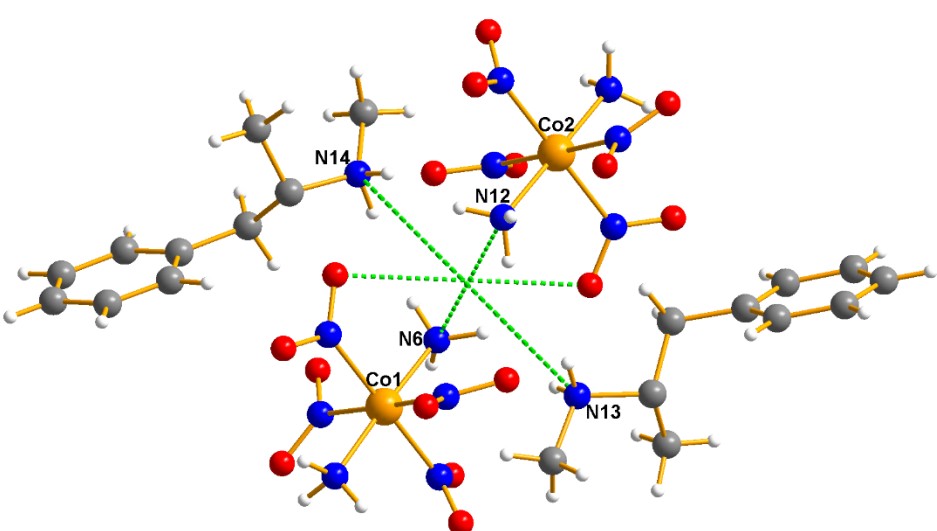

**Figure 10.** The pair of cations and anions observed in the case of methamphetamine. The intersection of the dotted lines is located at 0.4749, 0.5239, 0.5182, which is also very close to $\frac{1}{2}$, $\frac{1}{2}$, $\frac{1}{2}$, as expected for a case of a racemic mimic. Overlays, above, provide a rationale for the reason why both the cocaine and amphetamine cations can function thus.

### 3.3.3. π–π Contacts

The criterion for meaningful contacts between aromatic fragments labeled "π–π" interactions" in the report by Janiak [23] suggests that, given the experimental data available (see Figure 7 and relevant commentary in that paper), the range of 3.3–4.6 Å is reasonable. Using that as an acceptable gauge, our compounds do not have acceptable "contacts" in that range and should be ignored, because, in both cases considered here, they are closer to 6.0 Å. However, we think it is worth pointing out that that some meaningful "residual contacts" exist and depict them below in Figures 11 and 12. This caveat is in the same spirit as that in past discussions of the existence, or lack thereof, in hydrogen bonding discussions.

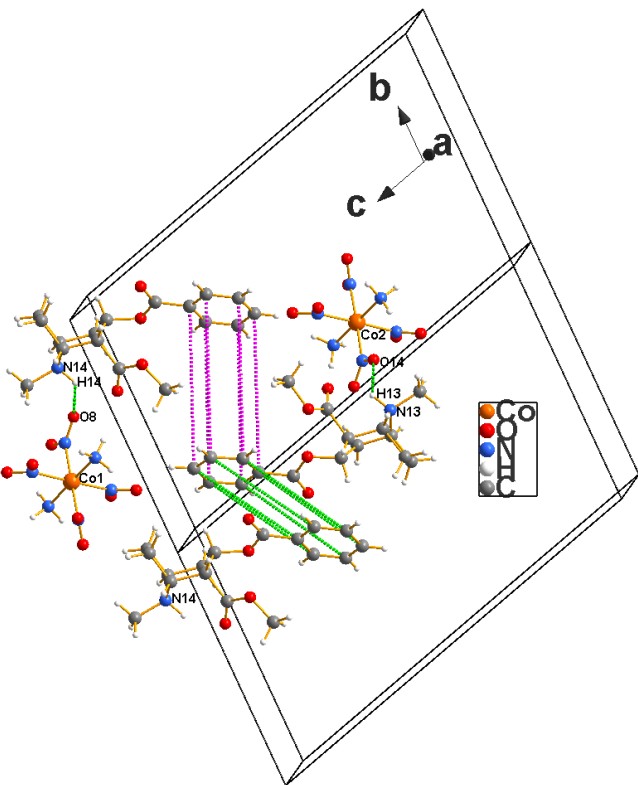

**Figure 11.** π–π interactions observed in the cocaine–Erdmann's complex crystals that are not separated by simple lattice translations, in which case the aromatic fragments in question are parallel to each other; thus, the latter are ideally suited for such electronic intermolecular interactions are and ignored here.

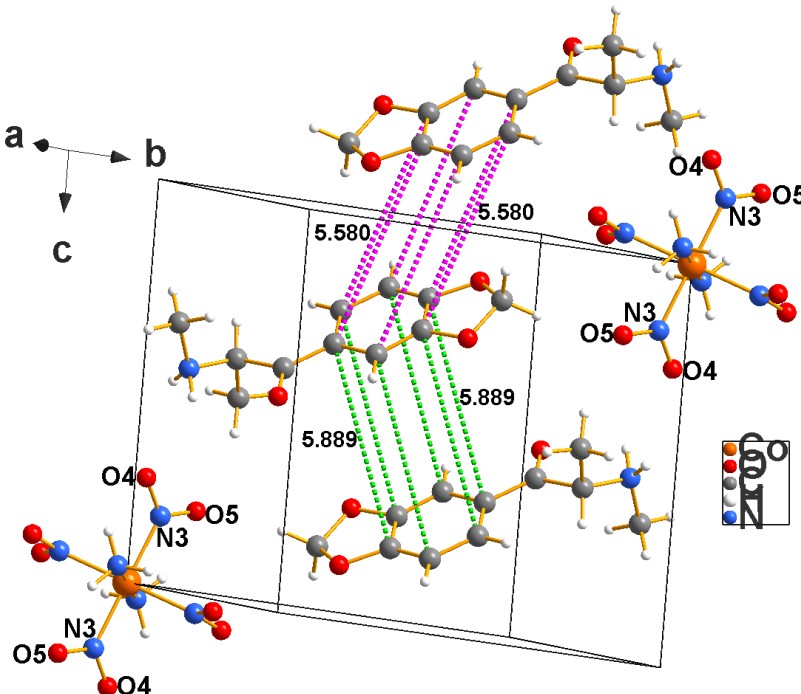

**Figure 12.** A packing diagram for the methylone complex with Erdmann's anion (**5**) is shown above. The same comments about π–π interactions in the case of cocaine apply here.

a.   Cocaine Cation with Erdmann's Anion (**3**)

The distances between the central ring (C10–C15) and the closest ring (C27′–C32′) range between 5.78 and 5.94 Å. The second ring atoms are generated through symmetry [x, y, z + 1]. The centroid-to-centroid distance = 5.868 Å [symm = x, y, z + 1], and the angle between the ring normal and the centroid-to-centroid vector is = 85.1° for this pair.

The other close π–π interaction is the central ring to the upper ring in the diagram, generated through symmetry [x, y, z + 1]; these distances range between 6.79 and 6.97 Å. The centroid-to-centroid distance = 6.868 Å [symm = x, y − 1, z], and the angle between the ring normal and the centroid-to-centroid vector is 51.7° for this one.

b.   Methamphetamine Cation with Erdmann's Anion

In crystals of methamphetamine cation with Erdmann's anion (**4**), there are no close π–π interactions other than those dictated by translations; thus, we saw no need to illustrate those in this instance.

c.   Methylone Cation with Erdmann's Anion (**5**)

The distances between the central ring (C10–C15) and the closest ring (C10′–C15′) range between 5.86 and 5.92 Å. The second ring atoms are generated through symmetry [1 − x, 1 − y, 1 − z]. The centroid-to-centroid distance = 5.889 Å [symm = 1 − x, 1 − y,1 − z], and the angle between the ring normal and the centroid-to-centroid vector is = 54.7° for this one.

The other close π–π interaction is the central ring to the upper ring in the diagram, generated through symmetry [1 − x, −1 − y, −z]. These distances range between 5.58 and 5.62 Å. The centroid-to-centroid distance is 5.889 Å [symm = 1 − x, −1 − y, −z], and the angle between the ring normal and the centroid-to-centroid vector is 49.5° for this one.

In conclusion, it seems that, whenever aromatic bearing fragments are not sterically hindered, π–π interactions, other than those dictated by lattice translations short enough to be meaningful, are useful in forming sturdier lattices such as observed in the cases of (**3**) and (**5**).

## 4. Summary of Experimental Results

The adoption of, and continued investigation of, the utility of Erdmann's salt in forensic analytical testing schemes will aid the analyst by reducing sample preparation time, as well as reduced reagent cost, and will allow easy transfer of a sample to a confirmation technique, such as infrared spectroscopy or X-ray powder diffraction. Finally, testing with Erdmann's salt is essentially a nondestructive testing technique, preserving the resulting precipitate for further analysis or courtroom presentation.

The powder diffraction patterns of the precipitation products of the potassium Erdmann's salt with each of the street drugs described here (structures (**3**), (**4**), (**5**)) (Figures S1–S3) will provide the necessary analytical confirmation to any forensic lab that is using powder X-ray diffraction in conjunction with crystal tests. These powder patterns are calculated from the single crystal structures using the powder pattern generating routine in the SHELX package. X-ray diffraction is considered a "Category A" technique by the Scientific Working Group for the Analysis of Seized Drugs (SWGDrug), due to its high discrimination capabilities [24].

## 5. Conclusions

It seems we have justified the use of Erdmann's anion as a valuable, easily accessible, and inexpensive reagent for the purification and co-crystallization of samples of illicit drugs confiscated in the streets or police raids. Large amounts of the ammonium and/or potassium salts can be prepared by very simple methods described above; the advantage of such a procedure is that a single, purified, large source can then be used for future forensic studies with the confidence of uniformity.

In the cases of crystallographic tests (single crystal or powder), the resulting test specimens we tested have been very satisfactory, especially when the results are subjected to the Flack Parameter test.

**Supplementary Materials:** The supplementary materials are available online at https://www.mdpi.com/article/10.3390/chemistry3020042/s1.

**Author Contributions:** S.M. was an undergraduate student who prepared the ammonium and potassium salts of Erdmann's complexes under direct supervision of R.A.L. M.R.W. obtained the drug samples, and he prepared the drug complexes under direct supervision of R.A.L., I.B., R.A.L. and M.R.W. together wrote the manuscript. All authors have read and agreed to the published version of the manuscript.

**Funding:** This research received no external funding.

**Data Availability Statement:** All data are deposited in CCDC.

**Acknowledgments:** We acknowledge the National Science Foundation for NSF–CRIF Grant No. 0443538 for part of the purchase of the X-ray diffractometer.

**Conflicts of Interest:** The authors have declared that no competing interests exist.

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
