# Peer review of "Erdmann’s Anion—An Inexpensive and Useful Species for the Crystallization of Illicit Drugs after Street Confiscations†"

_chemistry, doi:10.3390/chemistry3020042_

Round 1
Reviewer 1 Report
Dear Authors,
Thank you for an interesting manuscript. I recommend publication subject to revision and answers to the questions you will find in attachment with my comments and recommendations to improve your manuscript. These questions and comments do not signify major flaws in the study, but results could be presented more clearly (especially starting from Racemic mimics section).
Best Regards.

Author Response
The Submit below does not work. Here are our responses:
Reply to the Reviewer 1 Comments for Chemistry 1158222
General Statement:
We wrote a eulogy to Howard Flack in order to celebrate his life and highlight his scientific achievements, and not to display extensive command of the various topics we covered. In fact, we purposely chose material that allowed us to compactly show the breadth of crystallographic situations that the “Flack Parameter” serves to clarify and/or to document.
For that reason, we were forced to decide on the extent to which we would add and discuss published materials and their relation to our presentation, as well as presenting our own work as well. Therefore, assuming that professionals will be the ones reading the issue dedicated to Howard, we chose a variety of topics for which his work is of fundamental value, as opposed to a display of knowledge of the literature and the scholarly achievements of others; e.g., those in the suggested additional references. Practitioners are, no doubt, aware of those already.
Finally, we are grateful for all the literature information sent to us, which we will keep, and use, in forthcoming papers from our current investigations.
Here, we address the more specific issues brought up by Referee 1:
We are truly grateful for the very beautiful and scholarly information provided by this referee.
Concerning the SUMMARY:
Issue 1. Thanks, we needed that!
Issue 2. The origins, or most important contributions to the phenomenon of racemic mimics were traced to the papers cited in our manuscript. Among, and certainly most important and impressive, were those contributions which were parts of the studies that resulted in Odd Hassel winning the Nobel Prize. But, as requested, we have cited the significant summary of Herbstein’s contributions, as ref. 20 of the amended manuscript.
The General Comments (above) take care of the rest – we merely wanted to touch on the fact that Flack’s Parameter correctly identified this mode of crystallization as well.
Issue 3. See General Comments (above). This is not a proper forum to “debate” the Flack test. Here, we give credit to its usefulness, as it should be in an In Memoriam eulogy.
Concerning RECOMMENDATIONS:
Note about space group notation: Very silly comment, but we have complied to this request. Everyone reading these papers knows what the space group symbols are and what they mean.
About molecular overlay: The program MERCURY (a) attempts to overlay the two moieties as described in the CIF, and (b) attempts to optimize the fit. By selecting the second choice, we demonstrated that (1) if they still differ noticeably, the initial fit attempted must be worse, which is the point we wanted to make. Therefore, we presented our choice and figures.
Additional remarks:
Page 1, line 43: To our knowledge, NO references are allowed in any Journal’s Abstract.
Page 2, line 48: Took care of the missing period.
Page 3, Mat. & Meth.: the word “enantiopure” has been added.
Page 2, line 93: Title 2.1.2 has been put on the next page.
Page 3, line 108-109: GC/MS data not available.
Pages 3-5, Tables 1&2: All corrected.
Page 7, line 180-190: A caveat: two references about Flack’s work have been added.
Page 7, line 189: conglomerate crystallization is well-known phenomenon, as per attached documents (3 references are at the bottom of this remark).
Page 7, line 190: “ CD (circular dichroism) spelled out and added.
Page 8, line 208-213: conglomerate crystallization is well-known phenomenon, see above.
Page 10, line 262-264 In order to properly answer this, the reader should go to the literature.
Page 11, line 270: Done.
Page 13, line 313: A new Figure 11 has been inserted.
Page 14, line 330: A new Figure 12 has been inserted.
Page 15, line 379: Done.
Page 15, line 380-381: Done.
Page 15, line 395: Clevers & Coquerel reference added to Ref. 12 (on kryptoracemic crystallization).
References for conglomerate crystallization (from Pasteur on):
- Pasteur, Ann. Chim. Phys. 3rd Ser. (1850), 28, 56.
- Bernal, J. Chem. Educ. (1992), 69, 468.
- B. Kauffman, I. Bernal, H-W. Schütt, Enantiomer (1999), 4, 33.
Reviewer 2 Report
The work shows a good study concerning the characterization of street drugs via complexation. The structures are well characterized and the authors say that Flack parameters could solve the stereochemistry. But I have some points the authors must solve before publishing:
1) The π-π interactions are not well described in the text. The authors must improve the discussion on this part of the text (section 3.3.3) giving a more consistent discussion based on ring centroids parameters. Actually, it is very difficult to relate ring distances of 6.0 Å (described in the text for the e cocaine-Erdmann’s complex) to classic π-π interactions. Distances around 6.8 Å are clearly out of range for these interactions. Even ring distances around 5.7-5.8 Å (described in the Methylone cation with Erdmann’s anion) are in the upper limit for acceptable π-π interactions involving carbon atoms.
2) The authors simulated the powder diffraction of the complexes and say this is important to " provide the necessary analytical confirmation to any forensic lab ". I agree. Why the experimental powder diffraction is not presented in the paper - I think this should be done and compared to the simulation
Author Response
After Choosing the file, the Submit does not work. Here are our replies:
Reply to the Reviewer 2 Comments for Chemistry 1158222
1) “The π-π interactions are not well described in the text. The authors must improve the discussion on this part of the text (section 3.3.3) giving a more consistent discussion based on ring centroids parameters. Actually, it is very difficult to relate ring distances of 6.0 Å (described in the text for the e cocaine-Erdmann’s complex) to classic π-π interactions. Distances around 6.8 Å are clearly out of range for these interactions. Even ring distances around 5.7-5.8 Å (described in the Methylone cation with Erdmann’s anion) are in the upper limit for acceptable π-π interactions involving carbon atoms.”
Response: While writing the paper, we took this issue very seriously, as demonstrated below:
(a) as stated above, we tried to limit our presentation to topics for which the Flack parameter had some important contribution to make; but, we could not resist the introduction of the π-π interactions because these were widely present in street drugs. And, being aware of the Crystal Growth & Design Review, Vol. 20, May of 2019, we wrote to contributors regarding the limits of π-π interactions accepted by professionals; e,g., what are min-max values we should use with confidence.
(b) Here are the results of our inquiries:
(1) in the gas phase, π-π stacking of benzene occurs at 4.96 Å. Very informative, but useless in crystals. Moreover, we wanted a range of min-max values, which we did not get. Apparently, as in the case of hydrogen bonds, this is a topic long debated before any resolution was ever reached.
(2) we wrote to a couple of researchers from India who contributed to the above-mentioned document, and got no answer at all; probably due to the virus problems.
(3) Martinez, Chelsea R., Brent L. Iverson: “Rethinking the Term ‘pi-stacking’”. Chemical Science 3, nº 7 (2012): 2191–2201; doi:10.1039/C2SC20045G. We received a most interesting reply, which we summarize here:
There are two lines of thought concerning π-π interactions among theoreticians: those using a soft, broad potential, and those using a sharp potential function. They dislike each other. Iverson seemed to prefer the sharp potential approach as exemplified by a theoretician at University of North Carolina at Chapel Hill; but, watch out, he said. If we were to accept the views of one side, and get a referee from the other, we would probably be in trouble. Bottom line: excellent advice, BUT no numerical information at all.
Given a looming dateline, we abandoned that project which is not a subject we have much time for. Would you waste your time inquiring further into the subject? Please pick the side you prefer.
2) “The authors simulated the powder diffraction of the complexes and say this is important to ‘provide the necessary analytical confirmation to any forensic lab ‘. I agree. Why the experimental powder diffraction is not presented in the paper - I think this should be done and compared to the simulation.”
Reply: The experimental X-ray powder diffraction pattern is unnecessary, given that a more precise one can be generated from the CIF provided. We also provided in the Supplement a picture that can be used for comparison by those recording one of their own from drug seizures.
Round 2
Reviewer 1 Report
Dear Authors,
Thank you for some of your answers. I understand that the main point of this paper is not a review on the racemic mimics that is a very specific and restricted subject.
Nevertheless, numerous question I asked in the first report have received no answer. In the following, I strongly suggest some improvements to make your paper suitable for publication or at least give valuable answers to interrogations raised by your manuscript.
Best regards,

Author Response
Responses for Referee #1 (2nd round):
As far as the molecular overlays:
The way we created Figures 7 and 8, using MERCURY and DIAMOND:
1. Using the appropriate CIF, each containing a pair of fragments to be compared, we began the process of overlaying the fragments. Note that we knew in advance that the specimens were pure chiral pharmaceutical-grade; thus, the two drugs were of identical chirality. Therefore, all output relating to chiral inversions, etc. was ignored. Then, we
(a) Selected one atom (any one will do) from each fragment.
(b) Used the command CALCULATE and selected molecular overlay.
(c) A figure appeared on the screen, which is the EXACT overlay of the images as described by the coordinates in the CIF, and approximately resembling what is on Figures 7 and 8.
(d) The choice of output we selected was not the first attemp and image, but, the one displaying the best fit between the two of them.
Additional Explanation: Go to Figure 7 and note that atoms O20 and O24 are chemically related. So are O17 and O21, and C10 and C33. MERCURY understands that because it has something called “A connectivity diagram”. So, the next step is to minimize the distances between chemically-related atoms. It does that, one at a time, AND gives the second image: these are ones displayed in Figures 7 and 8.
Why did we choose these? Answer: Because, esthetically, they are a lot nicer than the first choice, AND show sufficiently well that there are large enough stereochemical differences to justify the existence of Z’ = 2.0 in the crystal. Even when optimized by least-squares. Choice of DIAMOND for the final figures:
DIAMOND is a far superior graphics platform than is MERCURY, whose function is more utilitarian, and it is our choice to make, anyway.
______________________________________________________________________________
As far as the racemic mimics are concerned:
We have reworded our paper to read the following:
3.3.2. Racemic Mimics.
Historically, it appears that an awareness of the existence of this type of crystalline material was first published in papers by a) Furberg and Hassel, who studied the crystal structure of phenyl glyceric acid slowly grown from water [16]; b) by Schouwstra, who studied crystals of DL-methylsuccinic acid grown by sublimation [17] and from water solution [18]; and, c) by Mostad, who examined o-tyrosine crystals grown from methanol containing small amounts of ammonia, to increase its solubility [19]. In all those cases, crystals of the racemate and of the optically pure material crystallized with identical cell constants; this leads to values of Z’ = 1 for the racemic samples and Z’ = 2 for the pure enantiomorphs.
[Caveat: because some of those lattices contained racemic pairs and had Z’ = 2.0, the authors of those days [16-19] labeled them racemates. In fact, the proper term, nowadays, would be kryptoracemates; but because we do not want, at this stage, to branch out into that topic, a brief but suitable discussion of this issue is given in Supplementary Material 2, below. We thank the referee for bringing this issue to our attention.]
Given that the two lattices (kryptoracemates and Sohncke space groups), Furberg and Hassel [16] asked: “why, and how?” In a remarkably clear and simple answer, they indicated that the pure chiral material seemed to crystallize as if a twin resembling in its packing that of the true racemate: in other words as a “racemic twin”; thus, the name Racemic Mimics that later evolved. They also proposed that substances containing flexible (dissymmetric) fragments whose torsional barriers were low, would make ideal candidates for the existence of such a phenomenon, and they documented additional cases [16].
(The overlay diagrams shown in this document show the extent to which torsional differences are associated with the observed Z’ value of 2.0). That was a remarkably advanced concept for its day – and happens to conform to what we describe in our presentation, since we have two cases of racemic mimics in the cases of the cocaine derivative and of the methamphetamine derivative of Erdmann’s salts. For readers interested in more extended commentary on this and related topics, we recommend Herbstein’s authoritative compendium [20].
where we have added the reference to Herbstein’s compendium.
______________________________________________________________________________
AND, we have added a Supplement Part 2 (for those interested in reading more about racemic mimics):
Supplementary Material – Part 2
When the earliest observations on what became labeled as Racemic Mimics occurred (ca. 1950-1975; see refs. 14-17 in the text above), the phenomenon of kryptoracemic crystallization had not been noted and published. The initial announcement, and the coining of the word, did not come until 1995, when it was revealed (ref. 12 above.) Consequently, the authors of paper (ref. 17) labeled the crystals of (+/-)- o-thyrosine as racemic. The fact is that they really constitute a kryptoracemic pair, as demonstrated below.
Racemates in Chiral Lattices
1. (+/-)-o-Tyrosine. Space group = P21. DTYROS
Structure ordered, z = 4; therefore, it has a racemic pair of ordered enantiomers as the asymmetric unit. See: A. Mostad, C. Rømming and L. Tressum, Acta Chem. Scand., 1975, B29, 171-176. Therefore, this is a case of kryptoracemic crystallization. The pure enantiomer packs almost identically to this one.
Note the pseudo-inversion center between the carboxylates. Also note that C8 and C17 are enantiomers. Additional, examples of such behavior are given below.
Please compare that kryptoracemic packing (of the racemate) with the structure published by Mostad, Rømming and Tressum, noted above.
The packing is identical for both; also note the pseudo-inversion centers in the pure chiral form displayed below.
A more modern view of the above packing is shown next:
The two racemic molecules are color-coded for emphasis.
Similar comments can be made on the structures below. However, this is not a proper forum for additional discussions, which can become quite elaborate. The brief remarks below may be helpful to the interested reader.
2. (+/-)-erythro-phenylglyceric acid. Space group = P21.
Structure ordered, Z = 4; therefore, it has a pair of ordered enantiomers as the asymmetric unit. Also, see: (a) C. N. Riiber and E. Berner, Ber., 1917, 50, 893; (b) S. Furberg and O. Hassel, Acta Chem. Scand., 1950, 4, 1020; (c) M. Cesario, J. Guilhem, C. Pascard, A. Collet and J. Jacques, Nouv. J. Chim., 1978, 2, 343. (Another kryptoracemate.)
3. (+/-)-carvoxime. Space group: P21.
Structure ordered, Z = 4; therefore, it has a pair of ordered enantiomers as the asymmetric unit. See: Fulton, M., Baert, F., Fouret, Acta Cryst., 1979, B35, 683. This species forms a series of continuous solid solutions with "unbalanced” amounts of enantiomers. The degree of disorder depends on the actual composition. That of the nearly (1:1) is almost ordered. The same is the case with carboximebenzene, Baert, F., Fouret, R., Oonk, N. A. J., Kroon, J., Acta Cryst., 1978, B34, 222.
4. DL-methylsuccinic acid. Space group: P21.
This structure has Z = 4; it has a pair of ordered enantiomers as the asymmetric unit. See: Y. Schouwstra, Acta Cryst., 1973, B29, 1636. Therefore, it appears to be another kryptoracemate.
There are many other examples of such behavior in the case of organic, metallo-organic and coordination compounds, all of which constitutes a gigantic area of investigation; many are old structures of possible unreliable information, such as space groups. Therefore, this constitutes an enormous area of research on its own – not a topic to be a side-line of a paper such as this.
If the referee is really serious, please identify yourself and we can have a really long discussion of this topic. Maybe we can even publish some joint material on this subject.

Reviewer 2 Report
I can not find any pi-pi stacking distances greater than 6.0 A in solid state reported in the literature. Actually, normal pi-pi stacking in solid state are clearly smaller than 5.0 A (I was very condencent when cited 5.7 A in my last report, specially if considering the van der Waals radii of carbon atoms). The authors should cite literature references with cases of stacking interactions were centroid-centroid distances higher than 6.0 A occurs. If not, the discussion about these interactions must be taken off from the paper before acceptance. I also have to say that the discussion must cite centroid-centroid distances and and angles between planes, if the discussion concerning pi-pi stacking remain in the text after the authors present the references.
Additionally, concerning the powder diffractogram, I think that if the propose o the paper is to show the analytical viability of using the technique, the powder diffractogram from the street drugs must be coherent to the simulated one. Despite of this I think the authors answered well this part.
Author Response
Responses to Referee #2 Comments (2nd round):
We have revised our manuscript to read as follows:
3.3.3. Pi-Pi Contacts
The criterion for meaningful contacts between aromatic fragments labeled “π-π” interactions” in the report by Janiak [21] suggests that, given the experimental data available (see Figure 7 and relevant commentary in that paper), the range of 3.3-4.6 Å is reasonable. Using that as an acceptable gauge, our compounds do not have acceptable “contacts” in that range and should be ignored, because in both cases considered here, they are closer to 6.0 Å. However, we think it is worth pointing out that that some meaningful “residual contacts” exist and depict them below. This caveat is in the same spirit as that in past discussions of the existence, or lack thereof, in hydrogen bonding discussions.
For both structures, we have added the centroid-to-centroid distances and the angle between the ring normal and the centroid-to-centroid vector for each pair shown.